# Distal Tibial Hemimelia in Fetal Methotrexate Syndrome: A Case Study and Literature Review

**DOI:** 10.3390/children10020228

**Published:** 2023-01-27

**Authors:** Dae-Sik Jo, Seung-Hyun Lee

**Affiliations:** Department of Pediatrics, Wonkwang University College of Medicine, Iksan 54538, Republic of Korea

**Keywords:** methotrexate, fetal methotrexate syndrome, congenital malformation

## Abstract

Methotrexate (MTX), a folate antagonist, is used in various fields, including malignancies and rheumatoid or inflammatory autoimmune diseases. MTX is used for the non-surgical treatment of ectopic pregnancies and the elective termination of pregnancy. The teratogenic effects of MTX have been recognized since the 1960s. Fetal methotrexate syndrome (FMS) was established based on the study of congenital anomalies. Generally, there is a risk of FMS when MTX is used between four and six weeks after conception. Here, we reviewed the literature regarding MTX usage and described a case of FMS that was born with a rare anomaly, such as tibial hemimelia, in a mother who had received MTX 4 months before conception for the management of an ectopic pregnancy.

## 1. Introduction

Methotrexate (MTX), a methyl derivative of aminopterin, is a folic acid antagonist widely used as an antineoplastic agent. It is commonly clinically used in women of childbearing age for the treatment of autoimmune diseases, such as rheumatoid arthritis and systemic lupus erythematosus, and after the organ transplant [1]. In addition, it is also used with misoprostol for spontaneous abortion [2] and treating ectopic pregnancies [3,4].

The prevalence of ectopic pregnancies remains stable every year, but a recent national survey found that the use of MTX for non-surgical treatment of ectopic pregnancies increased rapidly from 11.1% in 2002 to 35.1% in 2007 [5]. However, MTX should be used with caution, despite the fact that a medical approach is being adopted in consideration of the risks and expenses of surgical therapy.

The teratogenic effects of MTX have been recognized since the 1960s [6]. Microcephaly, craniosynostosis, tetralogy of Fallot, pulmonary valve atresia, limb reduction defects, and syndactyly are known to have higher rates than other anomalies. Fetal methotrexate syndrome (FMS) was established based on the study of congenital anomalies [7].

Generally, there is a risk of FMS when MTX is used between four and six weeks after conception. Here, we present a case of a neonate born to a mother who had received MTX for treating ectopic pregnancy 4 months before conception and showed unusual skeletal anomalies. In addition, we have summarized many studies about FMS. 

## 2. Detailed Case Description

We present a newborn male infant who was delivered via elective cesarean section due to breech presentation with a gestational age of 38 weeks and 6 days, a weight of 3530 g (67 percentile), height of 51 cm (65 percentile), and Apgar scores of 9 and 10 at 1 min and 5 min, respectively. Decreased muscle tone and a wide anterior fontanelle were not observed, which are generally observed in FMS newborns. However, congenital anomalies were observed, such as a skin pouch on the right tibial area, club feet (Figure 1), oligodactyly on the right first toe and left first and second toes, syndactyly on the third and fourth toes (Figure 2), and radial polydactyly on the left hand (Figure 3). Other deformities, such as cranial dysplasia, facial anomalies, cryptorchism, hypospadias, micropenis, and a bifid scrotum, were checked for, but they were not observed in this case. Along with the limb malformation, a single umbilical artery was identified, and the newborn was admitted to the neonatal intensive care unit. Blood tests performed at admission showed Hb 17.8 g/dL, Hct 51.8%, WBC 11,480/μL, Platelet 291,000/μL, C-reactive protein (CRP) 0.53 mg/L, pH 7.36, PCO_2_ 40 mmHg, PO_2_ 57 mmHg, Base Excess −2.8 mM/L, and HCO_3_^−^ 22.6 mM/L.

The infant’s mother was 37 years old, had a 0–0–1–0 obstetric history, and had previously used MTX (1 mg/kg/day) three times every other day as a treatment for ectopic pregnancy until 4 months before conception. The mother underwent a quadruple test at 12 weeks of gestation, in order to measure the levels of α-fetoprotein (AFP), unconjugated estriol (E3), human chorionic gonadotropin (hCG), and dimeric inhibin A (DIA); however, no specific results were found. Her prenatal ultrasound performed at 25 weeks of gestation revealed an atrial septal defect (ASD), club feet, and a single umbilical artery. Echocardiography (vivid S5, GE Healthcare, Korea) performed after birth revealed a medium-to-large-sized atrial septal defect (approximately 10–11 mm) and patent ductus arteriosus. There were no abnormal findings on brain imaging, including ultrasound (vivid S5, GE Healthcare, Korea) and MRI (Ingenia Elition, Ingenia Cx, Achieva, Philips, the Netherlands). Peripheral blood chromosomal analysis and chromosomal microarray tests performed on the seventh day after birth were normal. Urinary tract sonography was conducted to confirm renal agenesis and hydronephrosis based on previously reported cases, but no specific results were obtained.

The infant was discharged on the 10th day after birth, and the follow-up and treatment of his cardiac and musculoskeletal diseases was continued in an outpatient setting.

## 3. Discussion

MTX is used for the non-surgical treatment of ectopic pregnancies and elective termination of pregnancy [8]. Its mechanism of action is that it inhibits dihydrofolate reductase (DHFR), a folic acid synthase, which lowers the availability of tetrahydrofolate. This has antiproliferative effects that prevent DNA synthesis and cellular replication, as tetrahydrofolate is a crucial cofactor in the production of thymidylate and de novo purine [1,7].

However, MTX is recognized to be a strong teratogenic agent when administered in large dosages to women for treating cancer or ectopic pregnancy [9]. Although MTX is generally used with caution when treating women of childbearing age, cases of accidental exposure during the treatment of underlying diseases, treatment for ectopic pregnancies, or unsuccessful medical abortions have all resulted in reports of various abnormal findings in the fetus. During the early stage of pregnancy, exposure to MTX can cause fetal death, and surviving fetuses may experience FMS, or ‘methotrexate embryopathy,’ resulting in limb defects, skeletal malformations, and cerebral anomalies [1,10]. It can also increase the risk of tetralogy of Fallot, a complete atrioventricular canal, growth restriction, and, in some cases, developmental delay in the fetus [11].

In the 1970s, animal experiments (such as on mice, rats, rabbits, etc.) for MTX were conducted, but most of the results were for MTX dose and teratogenicity, and the study of FMS seen in newborns was mainly conducted in the form of a review by collecting cases [1]. A previous study listed the most common disorders that can affect each body organ in FMS, ranked by the likelihood of incidence [7]. Facial deformities were the most frequently seen, such as low-set ears, retro-/micrognathia, hypertelorism, hypoplastic orbits, prominent eyes, epicanthal folds, auricular malformations, a broad nasal bridge, cleft palate, and a high arched palate [12,13]. 

The next most common anomalies were observed in the limbs, which include one or more mesomelia, clinodactyly of the finger(s), hypoplasia/absence of fingers and/or toes, clinodactyly of the finger(s), and an aberrant palmar crease. Other possible conditions include missing metatarsal or metacarpal bones, syndactyly, clubfoot, and long fingers [7]. In our case, the infant had a club foot, syndactyly on both the third and fourth toes, and an atypical thumb. Additionally, tibial hemimelia could be observed in the right tibial region via the radiograph. This is distinct from previously documented limb deformities.

Cranial anomalies are also frequently reported, including craniosynostosis, large fontanelles, skull defects, and hypoplasia of the skull bones. Since microcephaly, holoprosencephaly, choroid plexus cysts, cerebellar hypoplasia, and absent corpus callosum [7,14,15] are most frequently mentioned as central nervous system malformations, in the case of newborns with suspected FMS, additional examination of the central nervous system should be considered along with checking for conspicuous abnormal findings. In this case, both the brain ultrasonography and MRI revealed no unusual findings.

In one previous case, MTX was administered 4 weeks after a misdiagnosis of ectopic pregnancy, resulting in a tetralogy of Fallot. In addition, ventricular septal defects and atrial septal defects have also been reported [9,16,17]. In this case, a moderate-sized atrial septal defect was found on the infant’s echocardiography.

Although cases are limited, cryptochidism, hypospadias, micropenis, bifid scrotum, and ambiguous genitalia have been reported as MTX-induced male genital malformations, and the need for systematic evaluation of the genitourinary tract has been mentioned [11,15,18,19].

Chromosomal testing is generally considered when patients with multiple congenital anomalies are born. A balanced reciprocal translocation between de novo chromosomes 5 and 20 (46,XY,t(5:20)(q15;p12)) was once discovered; however, the majority of FMS infants had normal karyotypes [7].

Even when a genetic test is normal, the various congenital abnormalities listed above can all be used to diagnose FMS, along with a history of exposure to MTX doses over the threshold that induces teratogenicity at approximately 5–6 weeks of conception [11]. As a result, it is important to double-check the mother’s MTX dosage and timing. The MTX exposure time for inducing teratogenicity varied between 0 and 23 weeks after conception, according to prior studies. However, the key period for teratogenicity was identified as 6–8 weeks after conception when examining exposure to a single dose [20,21]. 

There are cases of newborns being delivered with a single umbilical artery, a hypoplastic heart, nonexistent pulmonary arteries, and hypoplastic lungs, after low-dose MTX treatment for psoriasis 3–4 weeks after conception [22], and in one case, the use of MTX for treating an ectopic pregnancy at 5 weeks of gestation led to the discovery of numerous malformations, including cardiac defects (periventricular septal defect, foramen ovale, and elevated pulmonary arterial pressure), multiple skeletal malformations, and ambiguous genitalia [18,23]. In addition, unlike previous studies suggesting that MTX doses higher than 10 mg per week can cause malformations [24], FMS was observed at a low dose of 7.5 mg per week [25].

Alterations in renal clearance are unlikely to have a substantial impact because most of the MTX is eliminated via urine without experiencing any notable changes [25]. However, MTX is maintained for several weeks in various tissues and cells, and the approximate half-life of MTX in red blood cells is 2 to 11.3 weeks [26]. The persistence of the drug in fetal tissues may be related to the long half-life of MTX [27]. MTX pharmacokinetics and pharmacodynamics may also be impacted by a variety of additional factors, including genetic factors, which could then have an impact on MTX sensitivity [7]. An international panel of rheumatologists advised stopping the use of MTX for at least 3 months before conception, in light of the finding that it can linger in the body as glutamate complexes for several months, particularly in liver cells [28].

## 4. Conclusions

In many FMS studies, the anomalies seen in newborns have been summarized as statistically frequent anomalies by integrating existing cases. Since these cases did not include healthy infants born after MTX exposure during pregnancy, it is difficult to directly evaluate the occurrence of methotrexate teratogenicity.

We reviewed the literature regarding MTX usage and described a case of FMS that was born with a rare anomaly, such as tibial hemimelia, in a mother who had received MTX 4 months before conception for the management of an ectopic pregnancy. MTX was administered to the mother in this case in accordance with generally accepted recommendations; however, the newborn was born with FMS.

The variability of MTX teratogenicity can be varied due to the dose, route, and timing of MTX administered to pregnant women and differences in the sensitivity of fetal tissues to MTX. Therefore, it is considered that additional research on FMS is needed, along with a detailed report on these factors.

## Figures and Tables

**Figure 1 children-10-00228-f001:**
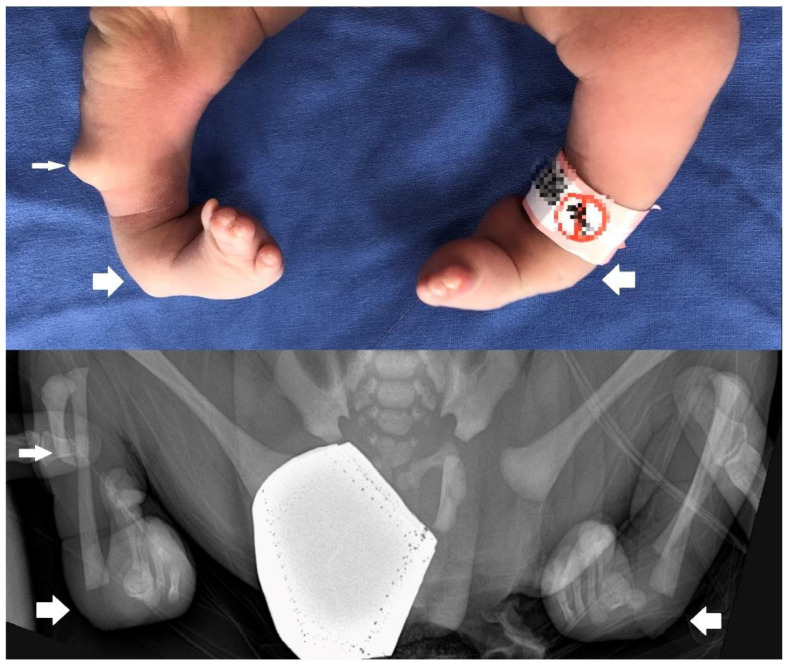
Lower extremity abnormalities, such as a skin pouch on the right tibial area (small arrow) and club foot (large arrows), are also present. X-ray of the lower extremity reveals abnormal right tibial hypoplasia, shown below (medium arrow).

**Figure 2 children-10-00228-f002:**
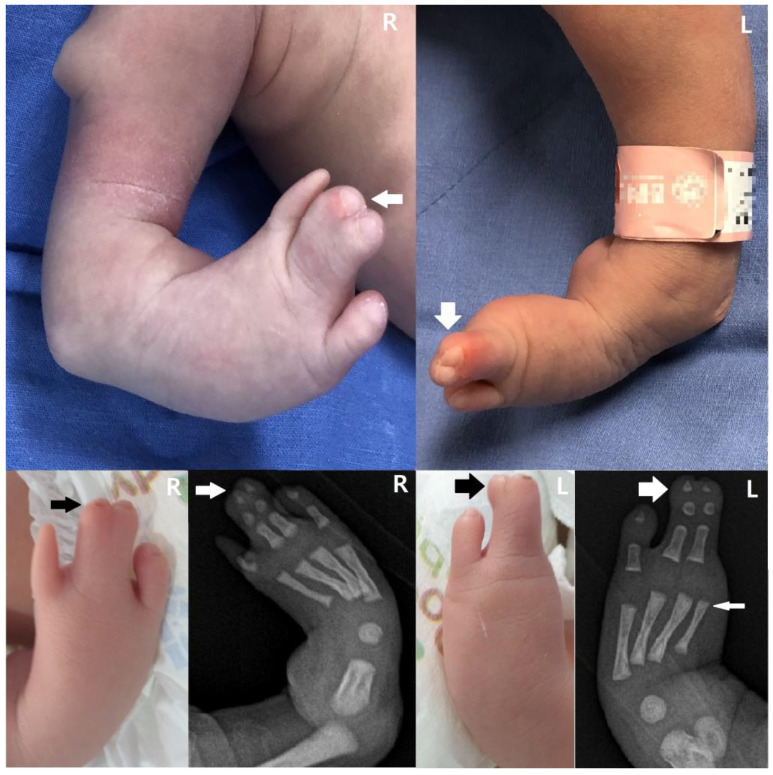
Oligodactyly of the right 1st toe and syndactyly on right 3rd and 4th toes of the right foot (medium arrows). Oligodactyly on left 1st and 2nd toes and syndactyly on left 3rd and 4th toes of the left foot (large arrows). X-ray shows that the left 2nd toe’s metatarsal bone is present (small arrow).

**Figure 3 children-10-00228-f003:**
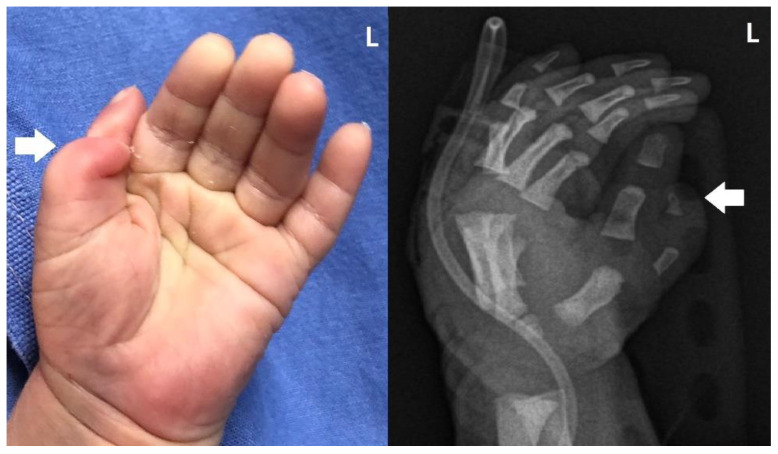
Radial polydactyly of the left hand (large arrows).

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
