# Peer review of "Distal Tibial Hemimelia in Fetal Methotrexate Syndrome: A Case Study and Literature Review"

_children, 2023, doi:10.3390/children10020228_

Round 1

Reviewer 1 Report

This is an interesting case, but there are several things I need to clear out:

1. What was the indication for performing cesarean delivery in this case?

2. Regarding the known risk of MTX exposure before conception, did you check the folic acid serum level? If yes, please provide us with the value

3. It would be so much more interesting if you included in your discussion about the effect of MTX on folate metabolism and the connection with embryogenesis/organogenesis

Author Response

Point 1: 1.What was the indication for performing cesarean delivery in this case?

Response 1: a ‘due to breech presentation’ is inserted in sentence. As a result, it was described as 'We present a newborn male infant who was delivered via elective cesarean section due to breech presentation with a gestational age of 38 weeks and 6 days...'.

Point 2: Regarding the known risk of MTX exposure before conception, did you check the folic acid serum level?

Response 2: Unfortunately, the mother's blood test results could not be confirmed. According to the information obtained from the mother through the questionnaire, she was taking folic acid regularly before pregnancy and that her folic acid level was not tested just before delivery.

Point 3: It would be so much more interesting if you included in your discussion about the effect of MTX on folate metabolism and the connection with embryogenesis/organogenesis.

Response 3: Due to the limited number of words, many clinical contents were summarized, so the contents of MTX and folic acid metabolism were briefly described in the first paragraph of the discussion. Unfortunately, no link between embryogenesis/organogenesis could be found. In the case of animal experiments, there were papers that mentioned MTX dosage and teratogenicity, but no references were found for embryonic or organogenesis.

Reviewer 2 Report

It is a very important report but the literature is very limited. It will be good to have this published to increase evidence base for future studies. And if they can add a little more in the literature on lab studies (if any). 

Author Response

Point 1: If they can add a little more in the literature on lab studies.

Response 1: As summarized in the review article, experimental studies were conducted in the 1970s with mice, rats, cats, rabbits, and monkeys. In the case of animal studies, there were papers that mentioned MTX dosage and teratogenicity.

The following sentence is inserted in the third paragraph of the discussion: "In the 1970s, animal experiments (such as mice, rats, and rabbits, etc.) for MTX were conducted, but most of the results were for MTX dose and teratogenicity, and FMS seen in newborns was mainly conducted in the form of review by collecting cases [1].”
